# Integrating Dynamic 3D Chromatin Architecture and Gene Expression Alterations Reveal Heterosis in *Brassica rapa*

**DOI:** 10.3390/ijms25052568

**Published:** 2024-02-22

**Authors:** Liu E, Shanwu Lyu, Yaolong Wang, Dong Xiao, Tongkun Liu, Xilin Hou, Ying Li, Changwei Zhang

**Affiliations:** 1National Key Laboratory of Crop Genetics & Germplasm Enhancement and Utilization, Nanjing Agricultural University, Nanjing 210095, China; 2021104078@stu.njau.edu.cn (L.E.); 2022104074@stu.njau.edu.cn (Y.W.); dong.xiao@njau.edu.cn (D.X.); liutk@njau.edu.cn (T.L.); hxl@njau.edu.cn (X.H.); yingli@njau.edu.cn (Y.L.); 2Key Laboratory of South China Agricultural Plant Molecular Analysis and Genetic Improvement, Guangdong Provincial Key Laboratory of Applied Botany, South China Botanical Garden, Chinese Academy of Sciences, Guangzhou 510650, China; shanwu.lyu@scbg.ac.cn

**Keywords:** heterosis, *Brassica rapa*, Hi-C, A/B compartments, TADs, integrative genomic analysis

## Abstract

Heterosis plays a significant role in enhancing variety, boosting yield, and raising economic value in crops, but the molecular mechanism is still unclear. We analyzed the transcriptomes and 3D genomes of a hybrid (F_1_) and its parents (w30 and 082). The analysis of the expression revealed a total of 485 specially expressed genes (SEGs), 173 differentially expressed genes (DEGs) above the parental expression level, more actively expressed genes, and up-regulated DEGs in the F_1_. Further study revealed that the DEGs detected in the F_1_ and its parents were mainly involved in the response to auxin, plant hormone signal transduction, DNA metabolic process, purine metabolism, starch, and sucrose metabolism, which suggested that these biological processes may play a crucial role in the heterosis of *Brassica rapa*. The analysis of 3D genome data revealed that hybrid F_1_ plants tend to contain more transcriptionally active A chromatin compartments after hybridization. Supplementaryly, the F_1_ had a smaller TAD (topologically associated domain) genome length, but the number was the highest, and the expression change in activated TAD was higher than that of repressed TAD. More specific TAD boundaries were detected between the parents and F_1_. Subsequently, 140 DEGs with genomic structural variants were selected as potential candidate genes. We found two DEGs with consistent expression changes in A/B compartments and TADs. Our findings suggested that genomic structural variants, such as TADs and A/B chromatin compartments, may affect gene expression and contribute to heterosis in *Brassica rapa*. This study provides further insight into the molecular mechanism of heterosis in *Brassica rapa*.

## 1. Introduction

Heterosis is a widely observed biological phenomenon whereby hybrid offspring exhibit superior traits compared to their parents, including growth potential, viability, yield, and quality [1,2,3]. Given the increasing global population, the successful application of heterosis in agricultural production dramatically contributed to augmenting crop yields, mitigating food crises, and guaranteeing food security [4]. The concept of heterosis has been applied for more than 100 years [5]. At present, three hypotheses, namely dominance [6,7,8,9], over-dominance [9,10], and epistasis [11,12], were proposed to explain the genetic basis of heterosis in most crops. However, these hypotheses remain theoretical and cannot fully explain the phenomenon of heterosis. The QTL mapping methods developed in the late 1980s laid a foundation for exploring heterosis, and extensive genetic analyses of hybrids in maize, rice, rape, and other plants led to the identification of numerous quantitative trait loci [13]. With the development of high-throughput sequencing technology, a series of dominant genes related to heterosis traits were identified by integrating transcriptomics and other omics [14], thereby offering new insights into the molecular mechanisms underlying heterosis. 

Genes encode genetic information in a one-dimensional linear sequence, which is expressed by forming a three-dimensional chromatin architecture. In addition, chromatin states and genome activity regulators also affect gene expression. Chromatin conformation plays a crucial role in executing biological activities via many regulatory elements [15]. Global and local chromatin rearrangements may occur upon sensing environmental and developmental cues, along with gene transcription changes [16,17]. Gene expression and regulatory network changes are entangled with heterosis [3]. In the interspecific hybridization between *Arabidopsis thaliana* and *Arabidopsis lyata*, the chromosomes from *Arabidopsis thaliana* in the hybrid are more compact, occupying a smaller nuclear volume and interacting more within the chromosomes [18]. These changes in the intensity of chromosomal interactions can affect gene expression. Studies have shown that the dynamic three-dimensional chromatin structure of *Brassica napus* also helps express plant hormone-related genes in hybrid F_1_ plants [19]. Systematically studying the frequency of interactions between genomic regions has led to the discovery of multiple principles of three-dimensional genome organization. Chromosome organization can be dissected into multiple functional and structural domains [20]. The units from high to low are chromosomal compartments (CTs), active (euchromatin, A-type) and inactive (heterochromatin, B-type) chromatin compartments, topologically associated domain (TADs), chromatin loops (CLs) and chromatin fibers. TAD regulates gene replication and epigenetic modification locally. The boundary region of TAD is genetically rich and conserved. The disruption of the TAD boundary can severely affect the regulation of gene expression and even lead to the occurrence of diseases [21]. The 3D genome and gene expression have complex relationships, with rapid dynamic changes in genome structure and gene expression accompanying hybridization. Nevertheless, current research on heterosis is mostly limited to one-dimensional linear sequences. Little is known about the changes in 3D structure for heterosis traits in *Brassica rapa*.

From appearance, Chinese cabbage (AA genome, 2n = 2x = 20) can be roughly divided into heading and non-heading Chinese cabbage. They are both highly nutritious, commercially viable, and easy to cultivate. They also have many differences, such as nutrient contentions and stress resistance. To exploit their advantages and explore the hybrid heterosis mechanism, we used non-heading Chinese cabbage 082 (P1) as the female parent and w30 (P2) as the male parent to generate the hybrid F_1_. We conducted a comparative analysis of Hi-C (high throughput chromatin conformation capture) and RNA-seq (RNA-sequencing) data, identifying several genes with specifically and significantly different expression levels in hybrids, and we discussed their potential functions and relationships. We also compared the 3D spatial structure of TADs and A/B compartments between the parents and hybrid F_1_ and observed the conserved and altered chromatin structures. We further investigated the associations between gene expression and the 3D chromatin structure. This study will contribute to the understanding of heterosis in Chinese cabbage, providing insight into achieving higher yields and better stress resistance through hybrid breeding.

## 2. Results

### 2.1. Heterosis Is Remarkable in the Hybrid F_1_

A remarkable biomass heterosis was observed in the hybrid F_1_ (see Figure 1A). To facilitate a quantitative comparison of their differences, we determined their fresh and dry weights at 21 days after sowing (Figure 1B). The values of MPH (mid-parent heterosis) and HPH (high-parent heterosis) both exceeded 50% (Figure 1C,D), and, in terms of fresh weight, MPH was as high as 159.28%. In comparison, HPH was as high as 150.11% (see Figure 1C). The F_1_ exhibited a remarkable degree of heterosis. The obvious phenotypic differences suited the subsequent integrative genomic analysis.

### 2.2. More Genes Were Actively Expressed in the Hybrid F_1_

High-quality transcriptome reference genomes of 124.57 Mb, 133.47 Mb, and 134.98 Mb were obtained for 082, w30, and F_1_ (Appendix A). Genes with FPKM ≥ 1 were defined as actively expressed. The number of actively expressed genes in the hybrid F_1_ (24,601) was higher than those in the maternal line 082 (24,320) and paternal line w30 (24,524) (Appendix A). We speculate that the expression of more active genes drives the formation of heterosis. However, no distinct differences were observed based on the genome-wide FPKM expression profiles for the three varieties (Appendix A). A pairwise analysis of these actively expressed genes further confirmed that many genes (more than 92%) were commonly expressed between the parents and that the male parent exhibited greater specificity than the female parent (Figure 2C). F_1_ had more genes that were specifically expressed when compared to either parent alone (Figure 2D,E). More genes between F_1_ and P2 (23,226) shared common expression profiles (Figure 2E). Concurrently, we compared the correlation of all expressed genes with nine samples and found that the correlation coefficient between F_1_ and P2 (R^2^ = 0.919) was higher than that between F_1_ and P1 (R^2^ = 0.888) (Figure 2G). Interestingly, when compared with both parents, F_1_ displayed the least specific expression of genes (485) compared to P1 (921) and P2 (891) (Figure 2F). We hypothesized that 485 genes with specific expressions in F_1_ may be involved in heterosis.

### 2.3. Transcriptome Analysis of DEGs

We analyzed the expression patterns and expression levels of differentially expressed genes (DEGs) to investigate the differential expression profiles of the genes. Firstly, we conducted a cluster analysis of the expression patterns. Some of the genes showed the same expression pattern between the parents and F_1_. In contrast, others showed significant differences (Figure 3A), suggesting that these genes may be responsible for the heterosis of the F_1_. Next, we analyzed DEG expression levels. A total of 2156 genes displayed significantly differential expression levels between the parents (*p* < 0.05, |log2FC| > 1) (Appendix A, Figure 3B, and Table 1), and the ratio between the maternal and paternal lines was similar. F_1_ and its parents differed significantly in 2568 genes, and most of those genes were expressed at higher levels in the hybrid F_1_ (Appendix A). Among these genes, we focused on 296 genes in F_1_ that had significantly different expression levels from both parents, and the expression level of most genes (173) in F_1_ was higher than that of its parents (Appendix A). We used the volcano map to summarize the significant DEGs (Figure 3B). These results further confirmed that hybridization activated the expression of more genes in F_1_ and made the expression level of these genes in F_1_ significantly higher than in its parents (Table 1). In addition, a Venn diagram analysis was conducted on the DEGs of the three groups (082 vs. w30, F_1_ vs. 082, F_1_ vs. w30), and 102 shared DEGs were identified (Appendix A). Among them, ten homologous hyperparental genes were identified in *Arabidopsis*, which were enriched in developmental growth, stress and resistance, response to light, the nitrogen compound biosynthetic process, the carboxylic acid metabolic process, and the lipid biosynthetic process (Appendix A). These genes may play an essential role in the growth, development, resistance, and other advantages of hybrid F_1_ plants.

### 2.4. GO and KEGG Analysis of DEGs

We conducted a gene ontology (GO) and KEGG pathway enrichment analysis to annotate the DEGs [22,23,24]. Using an over-represented *p*-value threshold of less than 0.018, we identified significant changes in the GO terms between the F_1_ and its parents. The top GO terms for biological processes were as follows: “response to auxin”, “nucleoside transport”, and “nicotinamide riboside transport”, while the majority of DEGs were found in “DNA metabolic process” (78 DEGs). In most studies, the metabolic process was unanimously recognized as related to growth or biomass heterosis [25,26,27]. In terms of cell components, the most common GO term was “Golgi-associated vesicle”, while “Cytoplasmic membrane-bound vesicle” had the highest number of DEGs (11 DEGs). The most common GO term for molecular function was “ADP binding” with the most DEGs (Appendix A and Figure 4A). We also identified the top 20 pathways assigned to our DEGs (Figure 4B). Most of the DEGs were involved in “Plant hormone signal transduction” (17 DEGs), “Purine metabolism” (17 DEGs), and “Starch and sucrose metabolism” (12 DEGs) (Appendix A). The “Plant hormone signal transduction” pathway helps the expression of plant hormone-related genes in F_1_, which is related to plant growth [19]. The main transport form of carbohydrates within plants is sucrose. The enrichment level of metabolic pathways in different tissues and development stages of different plants is different, and many studies suggested that carbohydrate metabolism is related to the formation of heterosis [24,28,29]. These findings suggest that hybridization results in a substantial proportion of transcriptome alterations, and more genes were activated than repressed (Appendix A). 

### 2.5. Genome-Wide Interaction Matrices of the Parents and F_1_

To explore the dynamic 3D spatial structural changes of the P1, P2, and F_1_ genomes during *B. rapa* hybridization, we conducted Hi-C sequencing and obtained 117.8 G, 141.1 G, and 133.0 G of raw data, respectively (Appendix A). We then mapped the clean Hi-C data to the high-quality *B. rapa* (v2.5) reference genome [30] and filtered out invalid pairs for subsequent comparative 3D structural analysis (Appendix A). The genome-wide simulation images showed that the P1, P2, and F_1_ nuclei were nearly spherical, and the chromosomes were localized in a limited volume (Figure 5A, Appendix A). The results confirmed previous findings that each chromosome occupies an exclusive region in the nucleus, a concept termed “chromosomal territory” [31]. At a resolution of 1 Mb, chromatin interaction analysis showed that the intra-chromosome interaction (*cis* interaction) of the three varieties was higher than the inter-chromosome interaction (*trans* interaction) (Appendix A). Chromosomal territory also revealed why intra-chromatin interactions were higher than inter-chromatin interactions. But F_1_ has a higher proportion of interchromosome/intrachromosome (*trans*/*cis*) contact than its parents (Appendix A). Therefore, each chromosome in the F_1_ nucleus may have a different space. To further explore the interaction patterns within chromosomes, we enhanced the resolution of the genome-wide interaction matrix to 100 kb (Appendix A). For example, among the interactions within chromosome 8 (Figure 5B), the dark red diagonal indicated the strongest interaction. Horizontal distance decreased the occurrence of intra-chromosome interactions. Supplementaryly, more refined components, like TADs (Figure 5C), were available. Each diagonally distributed triangle corresponded to a topological association domain (TAD) (Figure 5E). Meanwhile, there was no significant difference in the distribution of compartments (Figure 5D) between the F_1_ and parents.

### 2.6. Identification of A/B Compartment Shifts

To explore the changes in A/B compartments between the F_1_ and its parents, each chromosome’s A/B compartment distribution of the F_1_ and its parents was displayed (Figure 6A). The F_1_ contained more A compartments, while the parents tended to contain more B compartments (Appendix A and Figure 6B). A total of 473 genes from P1 to F_1_ were identified as A-to-B shifts, and 950 genes were identified as B-to-A shifts. A combined analysis with transcriptomics revealed 15 DEGs in A-to-B shifts (2 DEGs down-regulated in F_1_) and 20 DEGs in B-to-A shifts (16 DEGs up-regulated in F_1_). Meanwhile, a total of 281 genes from P2 to F_1_ were identified as A-to-B shifts, and 115 genes were B-to-A shifts. Moreover, 7 DEGs were in A-to-B shifts (1 DEGs down-regulated in F_1_) and 68 DEGs in B-to-A shifts (48 DEGs up-regulated in F_1_) (Appendix A). The inconsistent relationship resembled the results in Drosophila [32]. We found that hybridization can make it more inclined to transform into A compartments. In general, A compartments are in the transcriptional active region of euchromatin, and B compartments are in the transcriptional repressed region of heterochromatin. Therefore, the B-to-A compartment shift is considered beneficial to the formation of heterosis.

On this basis, we correlated the gene densities of 082-/w30-/F_1_ with A/B compartments. Results from F_1_ and its parents showed that the A compartment had a higher gene density than the B compartment (Appendix A). Further analysis of the gene expression of the A/B compartments found that there was no significant difference between the F_1_ and its parents. Nevertheless, the gene expression in the A compartment was significantly higher than that in the B compartment (Figure 6C). This was consistent with previous studies in *Arabidopsis* [33]. In summary, the A compartment, with its higher gene expression and higher gene density, may be a key factor in heterosis.

### 2.7. Changes in Compartments during Hybridization

During hybridization, 90.31% (A-to-A-to-A shifts are 72.96%, B-to-B-to-B shifts are 20.35%) of compartments did not change, suggesting that compartments in the nucleus are relatively stable. Furthermore, the F_1_ showed a higher proportion of compartment A, which increased by 4.04% (including 3.14% for A-to-B-to-A shifts and 0.90% for B-to-A-to-A shifts), while compartment B decreased by 1.44%, compared to the parents. Moreover, very few compartments underwent completely different transformations from their parent; of those that did, A-to-A-to-B shifts accounted for 0.06%, and B-to-B-to-A shifts accounted for 1.15% (Figure 2D). Heterochromatin remodeling is critical for a variety of cellular processes [34]. Therefore, the B-to-A compartment shift is considered beneficial to the formation of heterosis. Interestingly, 485 SEGs in the F_1_ showed similar results, but A-to-A-to-B shifts did not occur. When the parents’ compartments were not aligned, the compartments in the F_1_ were more inclined towards compartment A (Appendix A). The A/B compartment transformation analysis of 296 DEGs with significantly different expression levels in the parents and F_1_ was similar to that of the SEGs (Appendix A). These results suggested that different compartment activities between parents may influence compartment transitions in hybrids, ultimately impacting gene expression.

### 2.8. Identification of Different Kinds of Topologically Associating Domains

TADs, as relatively static physical and regulatory domains, facilitate specific and intentional gene expression programs in various ways [35,36]. To explore the 3D genomic differences between the parents and F_1_ more closely, we compared the change in TADs. The F_1_ shared similarities and differences with its parents regarding TADs (Appendix A). Further statistical analyses of the number and length of TADs showed that the number of TADs in the F_1_ was greater than that of its parents, and the average length of each TAD was smaller, as was the length of the whole genome (Appendix A and Appendix A). TADs are usually considered a relatively conservative structure, but the number and size of TADs change during cell differentiation. Therefore, we believe that changes in the number and size of TADs in hybrid F_1_ plants benefit heterosis. 

Compared with P1, F_1_ exhibited 46 specific TADs and 105 conserved TADs (12 activated TADs, 10 repressed TADs, and 83 other TADs) (Figure 7B). Similarly, compared with P2, there were 91 conserved TADs (10 activated TADs, 9 repressed TADs, and 72 other TADs) and 53 specific TADs in F_1_ (Figure 7B). A total of 141 DEGs were recognized in activated TADs (53 up-regulated DEGs in P1/P2), and 43 DEGs were identified in repressed TADs (42 down-regulated DEGs in P1/P2) (Appendix A). Together with transcriptome analysis, some DEGs were not expressed consistently with TAD types, and some DEGs were not detected in active or repressed TADs. These observations suggested that heterosis formation may be largely due to trans-regulatory mechanisms. Figure 7C showed that activated TADs had a percentage of fold change (P1 or P2/F_1_) >1 than repressed TADs. Moreover, the expression change in activated TADs was higher than that of repressed TADs (Figure 7D). The correlation analysis of different boundary regions was carried out based on the distribution of DI values in the upstream and downstream windows. We found more specific TAD boundaries between the parents and F_1_ and less specific TAD boundaries between the parents (Appendix A). This may be due to the large-scale rearrangement of chromatin during hybridization. Indeed, when the TAD boundary is disrupted or reconstructed, it can promote the formation of new promoter–enhancer interactions, thereby altering gene expression. These results indicated that the changes in 3D chromatin structure between the parents and F_1_ were greater than those between the parents. 

### 2.9. Identification of Candidate Genes for Heterosis and Verification of qRT-PCR

Based on the above transcriptome and 3D genome joint analysis results, we identified 140 DEGs associated with genomic structural alterations, including 56 DEGs in A/B compartments and 87 DEGs in TADs (Appendix A). Among them, three photosynthetic genes (*BraA02003217*, *BraA07001020*, and *BraA07001021*), eight plants cell size/division/cycle-related genes (*BraA06003576*, *BraA05002844*, *BraA07001159*, *BraA03006134*, *BraA05001791*, *BraA02002099*, *BraA02003420*, *BraA02002337)*, four carbohydrate metabolism genes (*BraA07001318*, *BraA06002248*, *BraA07001149*, *BraA01003773*), nine resistance- and stress-related genes (*BraA04000638*, *BraA07000080*, *BraA09003537*, *BraA09003725*, *BraA02003318*, *BraA07001184*, *BraA06002291*, *BraA07001349*, *BraA06002292*), three genes related to the response to auxin (*BraA02003420*, *BraA06000631*, *BraA07001182*), one development and cell death gene (*BraA05002866*), and one senescence-associated gene (*BraA08002232*) were identified. For example, the P1 vs. F_1_ group used F_1_ as a reference. If the TAD type of P1 relative to F_1_ was a repressed TAD, then P1 relative to F_1_ should be down-regulated. Notably, two candidate DEGs (*BraA02003296* and *BraA02003297*) were shared between these categories. To verify the accuracy of RNA-seq data, we selected 10 DEGs between the parents and F_1_ in 173 DEGs mentioned above by real-time quantitative PCR (qRT-RCR) (Appendix A). The expression patterns of these 10 DEGs were highly consistent with the data obtained by RNA-seq (Appendix A).

## 3. Discussion

The underlying molecular mechanism of heterosis remains elusive, and the use of three-dimensional structural dynamics on heterosis is still in its infancy. Therefore, integrating 3D genomics and transcriptomics is imperative for gaining a more comprehensive understanding of the role of chromatin in 3D space in heterosis.

Here, we present a study on the effects of hybridization on the 3D chromatin architecture and gene expression in *Brassica rapa*. The transcriptome data confirmed that hybridization leads to the activation of gene expression in F_1_ plants. A total of 485 SEGs, 173 DEGs, above parental expression level, more actively expressed genes, and up-regulated DEGs in F_1_ confirmed this conclusion. However, only a few genes were specifically expressed or displayed significantly different expression levels. These indicated that the genome-wide expression profiles of the hybrid F_1_ and parents were similar. Previous studies on heterosis produced by intraspecific hybridization also confirmed that most genes in hybrids and parents showed identical expression profiles, and the expression level of those genes was close to MPV [25,28,29]. In *Brassica napus*, Hu [19] also found that these hybrids had more up-regulated genes compared to the mid-parental value of gene expression, suggesting that altered gene expression patterns may influence their heterotic phenotypes. The DEGs detected in the F_1_ and its parents were mainly involved in the response to auxin, plant hormone signal transduction, the DNA metabolic process, purine metabolism, and starch and sucrose metabolism, suggesting that these biological processes may play a crucial role in the heterosis of *B. rapa*. However, the specific genes and pathways involved in heterosis may differ from species to species. In *Brassica napus*, the study found that hybrids with superior heterosis (better performance than both parents) had larger leaf sizes, which was attributed to increases in both cell size and cell number [19]. The differences in heterosis between species may be attributed to the specific genes and pathways involved, as well as the extent to which chromatin architecture influences gene expression. Further research is needed to fully elucidate the molecular mechanisms underlying heterosis and how they may vary between different *Brassica* species.

Although Hi-C technology has been widely used in many crops [37,38,39], little is known about genome architecture and its effect on *B. rapa* in heterosis. A preformed and stable topology (TAD) organizes the physical proximity between enhancers and their target genes. We compared the specific TAD and TAD boundary and the length and quantity of the TAD genomes. The results showed that the number of TADs in the F_1_ was greater than that of its parents, and the average length of each TAD was smaller, as was the length of the whole genome. This change results in the chromosomes of the hybrid being more compact than those of the parents, with higher interactions within the chromatin. Furthermore, we found that the F1 was more active than that between the chromosomes of both parents by calculating the interaction ratio of inter/intra. This may also be one of the reasons why the F1 had more active genes and up-regulated genes. Parents and hybrids exhibit significant differences in terms of TAD. More specific TAD boundaries between parents and the F_1_ were found. Changes in TAD boundaries can lead to gene rearrangement, thereby altering gene expression. It is precisely these changes in gene expression that give hybrid varieties heterosis traits. The chromatin structure of TADs is widely present in many crops, including rice and cotton [38,39]. However, there is no classical TAD interaction pattern in rice and *Arabidopsis*. Many species in the *Brassica* genus have been identified to have chromatin structures containing TADs, including *B. napus*, *B. rapa*, and *B. oleracea*. This study utilized three-dimensional spatial structural data to explore the impact of chromatin structural variation on heterosis traits, providing us with a new research perspective. 

Hybridization allows F_1_ plants more A compartments, resulting in more high-density and highly expressed genes. In *Brassica napus*, F_1_ hybrids with superior heterosis had more transcriptionally active A compartments in their chromatin compared to those with inferior heterosis [19]. Although there are no obvious differences in A/B compartments, the change in TADs may be one of the reasons why hybridization activates more genes in F_1_ plants. Subsequently, 140 DEGs with genomic structural variants were selected as potential candidate genes. Their functions were related to various processes, such as photosynthesis, plant cell size/division/cycle, resistance and stress, response to auxin, and carbohydrate metabolism. However, further functional verification is required for these candidate DEGs. In the future, we will obtain a mutant of the candidate DEGs through gene editing or other methods, observe its corresponding phenotype, and then obtain overexpressed plants to further determine the phenotype. In addition, the expression pattern analysis of the candidate DEGs is also needed (for example, analyzing expression levels in different tissues at different stages or expression sites in cells). If possible, we will further investigate interacting proteins and the regulatory relationships between upstream and downstream.

Gene expression is precisely regulated by the multi-layered three-dimensional structure of chromatin [36]. Different layers of the 3D genome have various levels of regulatory control [40]. This study specifically focused on the beneficial aspects of the 3D genome. However, further exploration is needed to understand the specific regulations. With the advancement of technology, it is believed that the impact of the spatial structure changes of chromatin on heterosis will be able to be further explored and interpreted in the future.

## 4. Materials and Methods

### 4.1. Plant Materials, Growth Conditions, and Sample Collection

The male parent w30 (P2), female parent 082 (P1), and their hybrid F_1_ used in this study were all provided by Nanjing Agricultural University (Nanjing, China). The F_1_ was generated by hand pollination. To reduce bias, all plant materials used for Hi-C and RNA-seq were grown in the same environment (24 °C, 16 h of light/8 h of darkness, 75% RH). Samples for Hi-C and RNA-seq were collected using fully expanded parental and hybrid F_1_ leaves (3rd upper leaf of a plant) with three replicates per sample. After being quick-frozen in liquid nitrogen, they were placed in a −80 °C refrigerator for later use.

### 4.2. Evaluation of Heterosis

To measure the phenotypic data, including the fresh weight (FW) and dry weight (DW), 30 similar plants (10 plants per replicate) were collected at 21 days. The collected material was washed with distilled water immediately and then baked at 105 °C for 30 min before further drying at 80 °C for 60 h [19]. MPH (mid-parent heterosis) was calculated as follows: MPH = (the mean value of hybrid F_1_—the average value of both parents)/the average value of both parents × 100%. HPH (high-parent heterosis) was calculated as HPH = (the mean value of hybrid F_1_—the optimal parental value)/the optimal parental value × 100%. 

### 4.3. RNA-Seq Experiment and Sequencing

Per the kit’s instructions, the total RNA of nine samples was extracted using the RNAprep Pure Plant Kit (Tiangen, Beijing, China). In addition, the construction and sequencing of cDNA libraries were also completed by the Novogene Company (Beijing, China). The construction of cDNA libraries included the synthesis, purification, and end repair of double-stranded cDNA, adding A-tail, ligation of sequencing adapter, PCR enrichment, and so on. After the completed libraries were qualified, different libraries were pooled according to the requirements of an effective concentration and target data volume, and finally, RNA sequencing was performed.

### 4.4. Hi-C Libraries Construction and Sequencing

The plant leaves were ground in liquid nitrogen, as described in Section 4.1 of this study. Then, 2 g of uniform powder was taken to establish a Hi-C library. The establishment of Hi-C libraries and completion of Illumina sequencing were conducted by the Novogene Bioinformatics technology company (Beijing, China). The libraries were built according to the previous standard protocol, with some modifications [41]. After fixing it with polyformaldehyde and DNA, restriction endonuclease was used to make a gap at the cross-linking point, and a biotin marker was used simultaneously. The adjacent DNA fragments were treated with nucleic acid ligase, then the protein at the junction was digested with protease. Finally, the fragments with a length of 350 bp were broken by the Covaris crusher for recovery. After the libraries were constructed, Qubit 2.0 and Q-PCR were used for preliminary and precise quantification, respectively, while Agilent 2100 (Agilent Technology Co., Ltd.; USA) was used to detect the insertion size of the library. Finally, the different libraries were pooled and sequenced once they had been qualified. 

### 4.5. RNA-Seq Data Analysis

Spliced reads were effectively compared to RNA-Seq data using HISAT 2 (version 2.1.0) [42]. In the RNA-Seq analysis, to make different genes and sequencing data comparable, we introduced the concept of FPKM. FPKM represents the number of fragments per million from a gene per kilobase length [43] and utilizes DESeq2 R (version 1.18.0) software for differences in gene analysis (DEG), selection criteria for |log2 fold change| > 1, or error rate (FDR) < 0.05. The corrected *p*-value was obtained by multiple hypothesis testing based on Benjamini and Hochberg’s method, and the square of the Pearson correlation coefficient between biological repetitions (R^2^ > 0.8) was used to test the experimental reliability. The Gene Ontology (GO) and Kyoto Encyclopedia of Genes and Genomes (KEGG) databases (www.kegg.jp/kegg/kegg1.html. accessed on 4 August 2023.) [22,23,24] were used to identify possible biological functions and pathways of DEGs. Among them, the GOseq (Release 2.12) software package was used to conduct GO enrichment analysis of the DEGs in R [44]. Cluster Profiler R package [45] was used to study the enrichment of DEGs in the KEGG pathway. 

### 4.6. Hi-C Read Mapping 

The quality of the obtained Hi-C sequencing data was controlled before output. Paired reads, including adapter-contaminated sequences, unknown nucleotide “N” ratios >10%, and more than 50% base Q < 5 were filtered out. Next, the qualified reads obtained were processed using the HiCUP pipeline (version 0.57) [46]. For data comparison, Bowtie2 software (version 2.2.3) [47] was used to compare the obtained reads with the *B. rapa* genome (version 2.5) [30]. The observed interaction matrices were constructed for the final influential contacts according to the statistical interaction matrices with a certain resolution interaction matrix. The maximum likelihood method was used to standardize the observation interaction matrix. The MDS algorithm of the PASTIS software (Release 0.3.3) [48] was employed to imitate the 3D position of chromatin. On this basis, the constructed heat map of chromatin interactions was divided into A/B compartments with a resolution of 100 kb by principal component analysis (PCA). TadLib (hitad 0.1.1-r1) software was applied to estimate the TAD topology, with default parameters at a 40 Kb resolution. RNA-seq was performed on the genes in TADs, and we classified the TADs into three categories based on the proportion of genes with positive and negative FC in each TAD of P1/P2 vs. F_1_: activated TADs, repressed TADs, and other TADs. We sorted the foldchange from largest to smallest, with top 10% being considered activated TADs, bottom 10% being expressed TADs, and the rest being considered others TADs. 

### 4.7. Quantitative Reverse-Transcription PCR 

QRT-PCR was used to perform RNA-seq verification on 10 randomly selected DEGs. In the qRT-PCR, cDNA was derived from the reverse transcription of three biologically repetitive RNAs of the parents and hybrids. Total RNA extraction was carried out according to the instructions of the RNAprep Plant Kit (DP419; Tiangen, Beijing, China). For reverse transcription, an Evo M-MLV RT Mix Kit with gDNA Clean [AG11728, Accurate Biotechnology Co., Ltd. Changsha, China] was used, the QuantStudio Q3 instrument was used for quantitative analysis, and Excel was used for mapping and data analysis. The primers of corresponding candidate DEGs are listed in Appendix A. BrActin (*Bra028615*) was used as a quantitative reference [49]. The 2^−ΔΔCt^ approach was applied to quantify the relative gene expression levels [50].

## 5. Conclusions

Our study revealed the gene expression profiles and changes in the 3D chromatin structure of *B. rapa* hybrids and their parents. Several gene regulatory networks may play a crucial role in the heterosis of *B. rapa*, including those involved in “plant hormone signal transduction” and “starch and sucrose metabolism”. The identification of 173 genes with higher expression in the F_1_ and 485 genes with specific expression in the F_1_ may contribute to the generation of heterosis traits. The differences in heterosis between species may be attributed to the specific genes and pathways involved, as well as the extent to which chromatin architecture influences gene expression. Further research is needed to fully elucidate the molecular mechanisms underlying heterosis and how they may vary between different Brassica species.

## Figures and Tables

**Figure 1 ijms-25-02568-f001:**
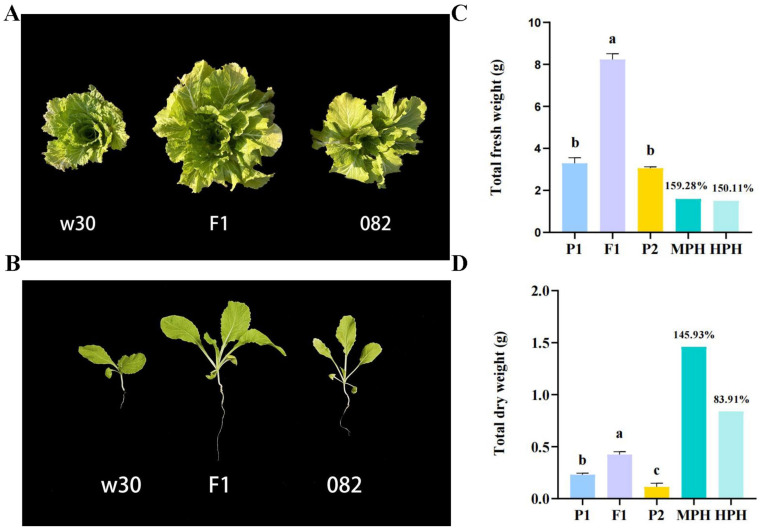
Comparisons of heterosis in w30(P2)-/082(P1)-/F_1_ traid. (**A**). Phenotypes of the hybrid F_1_ and its parents in *Brassica rapa*. (**B**). W30(P2)-/082(P1)-/F_1_ traid at 21 DAS. (**C**). The total fresh weight of the w30(P2)-/082(P1)-/F_1_ traid. (**D**). The total dry weight of the w30(P2)-/082(P1)-/F_1_ traid. MPH indicated the mid-parent heterosis in the w30(P2)-/082(P1)-/F_1_ traid, while HPH indicated over-parent heterosis. Values with different letters differ significantly (*p* < 0.05).

**Figure 2 ijms-25-02568-f002:**
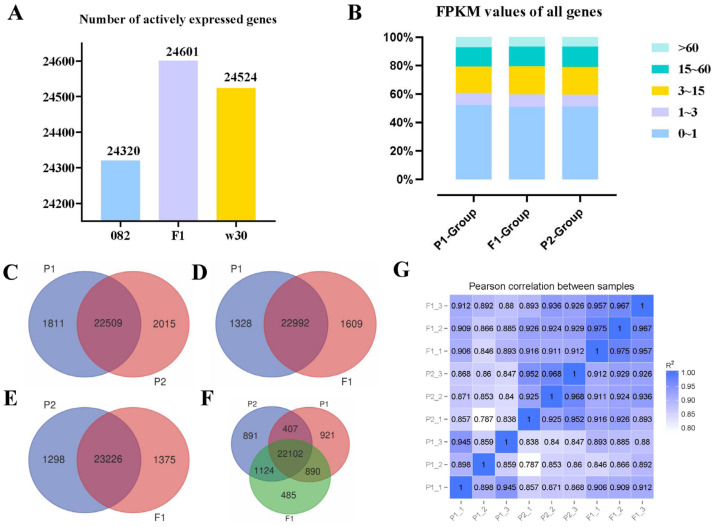
Comparative analysis of the actively expressed genes detected in w30(P2)-/082(P1)-/F_1_. (**A**). Number of actively expressed genes in F_1_ and its parents. (**B**). Genome-wide expression profile of w30(P2)-/082(P1)-/F_1_ traid. (**C**–**F**). Venn diagrams of the actively expressed genes drawn by comparative analysis. (**G**). The expression profiles of all detected genes were used to calculate the correlation of nine samples.

**Figure 3 ijms-25-02568-f003:**
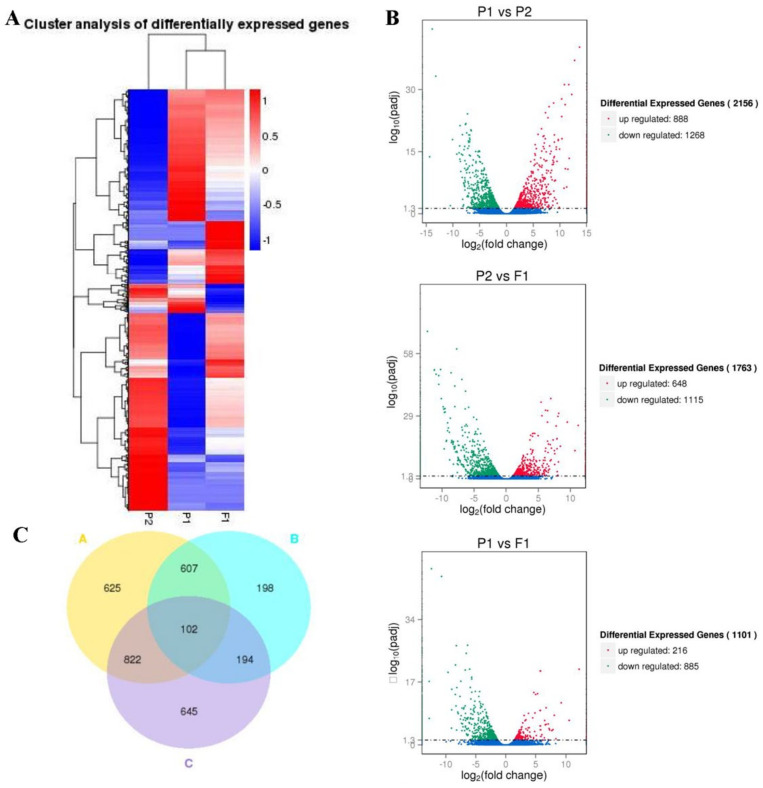
Overview of the DEGs of the hybrid F_1_ and its parents. (**A**). Cluster analysis of differentially expressed genes for w30(P2)−/082(P1)−/F_1_. The heat maps were constructed based on the log2 (fold−change in the normalized expression levels) of two arbitrary samples in w30(P2)−/082(P1)−/F_1_. The color scale represents the log2 (fold−change in the normalized expression levels) of two arbitrary samples, with blue denoting low expression and red denoting high expression. (**B**). Volcano map of DEGs. Red points denote up−regulated genes; green points denote down−regulated genes; blue points denote non−differentiated genes. Each group is referenced by the one after ‘vs.’, such as P1 vs. P2 is referenced by P2. (**C**). Venn diagrams of the differentially expressed genes were drawn by comparative analysis of the A, B, and C groups. Group A represents P1 vs. P2, group B represents P1 vs. F_1_, and group C represents P1 vs. P2.

**Figure 4 ijms-25-02568-f004:**
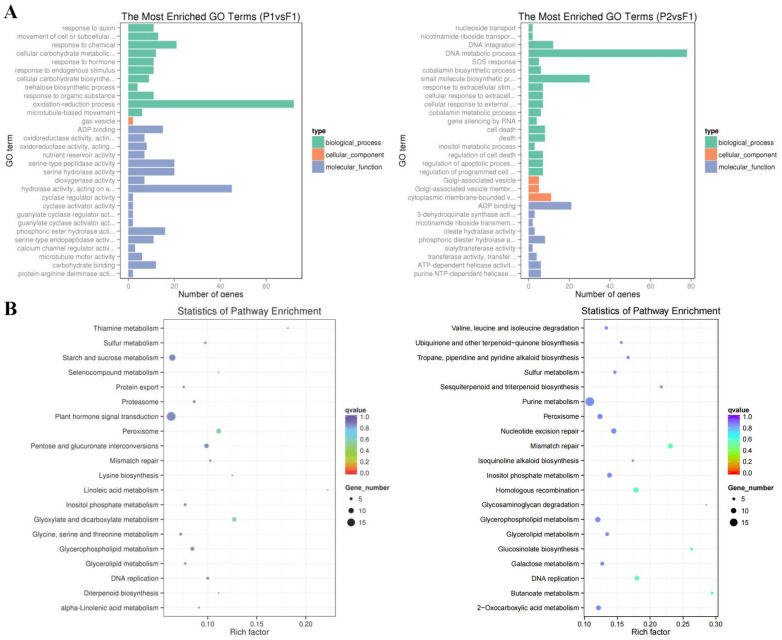
Analysis of diferentially expressed genes (DEGs) between the F_1_ and its parents. (**A**). The most enriched GO terms with all DEGs. The ordinate is the enriched GO term, and the abscissa is the number of DEGs. (**B**). The top 20 enriched KEGG pathways of DEGs. The pathway label is on the vertical axis, the Rich factor is on the horizontal axis, the size of the dots represents the number of DEGs in the pathway, and the color of the dots represents the distinct Q-value levels. Each group is referenced by the one after ‘vs.’; for example, F_1_ references P1 vs. F_1_. Kyoto Encyclopedia of Genes and Genomes (KEGG) databases (www.kegg.jp/kegg/kegg1.html. accessed on 4 August 2023.) were used.

**Figure 5 ijms-25-02568-f005:**
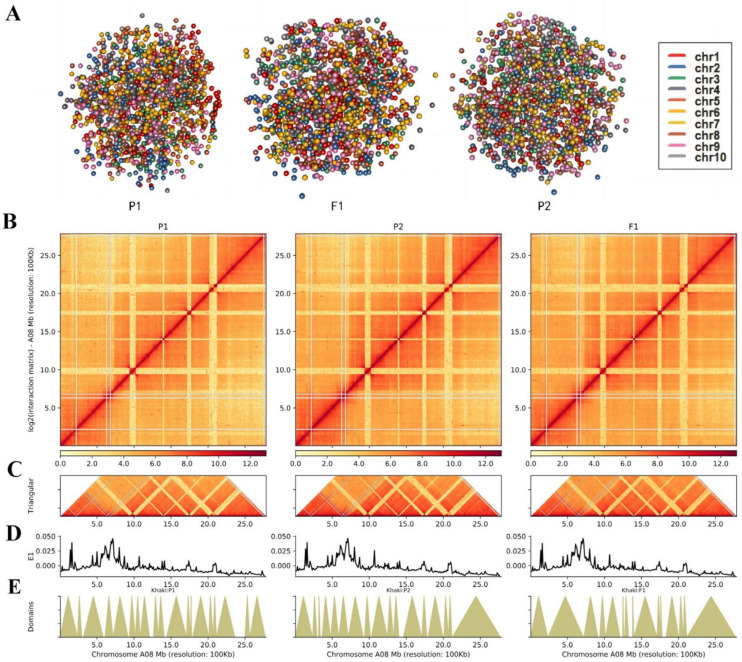
Hi-C analysis of chromatin contacts in F_1_ and parents on chromatin 8. The x-axis represents the chromosome. (**A**). Three-dimensional model of whole chromosomes. A different hue represents each chromosome. (**B**). Intrachromosomal interactions of the chromosome at 100 Kb resolution. The Y-axis represents log2 (interaction matrix). (**C**). Each triangle, distributed diagonally, is represented as a topologically associated domain (TAD). The y-axis represents Triangular. (**D**). The first principal component values show A/B compartment status. (**E**). Distribution of TADs. The y-axis represents domains.

**Figure 6 ijms-25-02568-f006:**
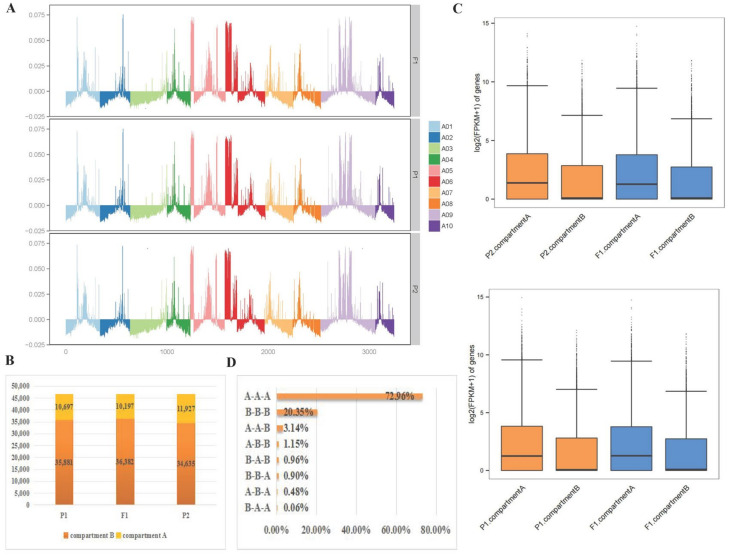
The analysis of the A/B compartments in the F_1_ and its parents. (**A**). The distributions of the A/B compartment on each chromosome. With the y−axis 0 scale line as the reference, above is Compartment A, and below is Compartment B. Different colors correspond to different chromosomes. (**B**). The number of genes contained in the A and B compartments. (**C**). Box plots representing the gene expression of the A and B compartments. The Y−axis represents log2 (FPKM + 1 of genes). (**D**). The percentage of the A/B compartment shifts between the F_1_ and its parents.

**Figure 7 ijms-25-02568-f007:**
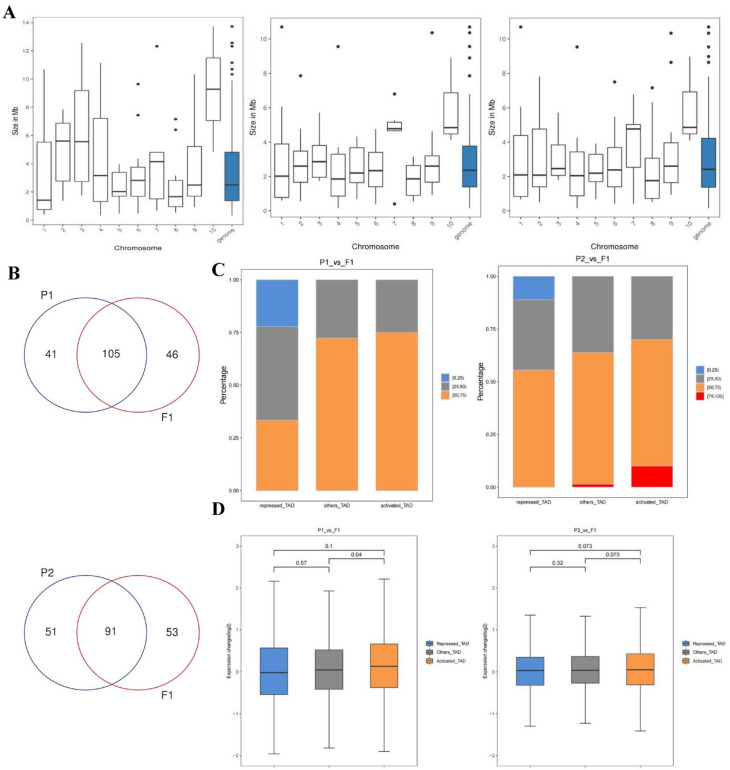
The analysis of the TADs in the F_1_ and parents. (**A**). The TAD length genomic distribution of the F_1_ and parents. From left to right: 082, F_1_, and w30. The Y−axis represents size in Mb. The X−axis represents the chromosome. (**B**). P1 vs. F_1_ Venn diagrams of the TAD. The overlapping part represents conservative TADs, and vice versa represents specific TADs. (**C**). Proportional stacking maps of genes with different expression levels in different TADs. For each gene contained in the TADs, the change in expression level was calculated for both sets of samples, and the percentage of all genes corresponding to that TAD with a fold change greater than 1. The percentages were divided into four categories, including 0–25%, 25–50%, 50–75%, and 75–100%. The results for the different categories of TADs were presented as bar charts. (**D**). Gene expression of the three types of TADs (repressed TADs/other TADs/activated TADs). The Y−axis represents expression change (log2).

**Table 1 ijms-25-02568-t001:** Differentially expressed genes detected in the hybrid F_1_ and their parents.

Hybrid Set Samples	Up	Down	B2P	Total
**082 vs. w30**	888 (41.19%)	1268 (58.81%)		2156
**F_1_ vs. 082**	885 (80.38%)	216 (19.62%)		1101
**F_1_ vs. w30**	1115 (63.24%)	648 (36.76%)		1763
**F_1_ vs. (082 and w30)**	173 (58.45%)	32 (10.81%)	91 (30.74%)	296

**Note:** DGPP indicated the differentially expressed genes (DEGs) between two parents. DGHP indicated the DEGs between the hybrid and parents. B2P indicated the expression levels of the DEGs between the two parental lines. Each group is referenced by the one after ‘vs.’; for example, 082 vs. w30 is referenced by w30.

## Data Availability

The datasets supporting the results of the present study are included within this article (and its Appendix A). All sequencing data generated for this study have been submitted to the NCBI Sequence Read Archive under accession number (PRJNA970959).

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
