# Peer review of "Integrating Dynamic 3D Chromatin Architecture and Gene Expression Alterations Reveal Heterosis in Brassica rapa"

_ijms, 2024, doi:10.3390/ijms25052568_

Round 1
Reviewer 1 Report
Comments and Suggestions for Authors
The manuscript "Integrating Dynamic 3D Chromatin Architecture and Gene Expression Alterations Reveal Heterosis in Brassica rapa" is a well-elaborated work that studies the nature of the heterosis effect from the perspective of chromosomal territories and the differential expression of genes by comparing two parental lines and their F1 progeny. The study design includes three biological replicates and three technical replicates. The conclusions are supported by the results and have both scientific significance and practical applications.
However, there are certain issues that should be addressed:
1) The resolution of all figures should be increased.
2) The diagrams in Figure 5 should be described in more detail. What is shown on the x and y axes in (b) and (c)? In lines 198-199, you wrote, "For example, among the interactions within chromosome 8." As the diagrams in Figure 5 show only chromosome 8, it should be indicated in the figure caption.
3) Figure 5a: Is it possible to attach a genuine 3D model in html in the Supplementary files? That would be truly spectacular.
4) Lines 346-355 should be replaced to the results.
In general, the text should be carefully checked for grammar and spelling (see Comments on the Quality of English Language).
Comments on the Quality of English LanguageIn abstract, "A" and "B" compartments and "B. rapa" should be deciphered
line 55 "Studies" should be in lowercase
line 73 "Hi-C and RNA-seq": should be deciphered
line 105 "the parents ," should be "the parents,"
lines 110-111 probably, "R" should be "R2" (check and verify)
In Figure captions, please, correct the style. "Indicated" after letters should be removed from captions of Figures, e.g. in lines 92, 93, 117, 118, 156, 281
lines 117, 281. 471, caption of Figure S9: "Venn diagrams" should be instead of "venn diagrams"
line 144 "NNote" should be "Note"
lines 193/194 "cis"/"trans" should be in italics
line 273 "Windows" should be in lowercase
line 212 "A/B Compartments": "Compartments" should be in lowercase
line 262 "Compared" should be in lowercase
line 264 "72other TADs": should be "72 other TADs"
line 299 "Real-time quantitative RT-PCR (qRT-RCR)": should be changed to "real-time quantitative PCR (qRT-RCR)"
line 419 "sequencings": should be changed to "sequencing"
line 436 "P value": "P" should be in lowercase
line 438 "R2 > 0.8": R2 should be R2
Figure S8 caption: "its parents samples" should be "its parents' samples"
Author Response
Responses to the editor and reviewers' comments
Dear editor and reviewers:
We are really thankful to you for the critical review and constructive comments.The manuscript has been revised accordingly. The revision in the MS is highlighted in yellow. Hopefully, the revised version of our manuscript ("Integrating dynamic 3D chromatin architecture and gene expression alterations reveal heterosis in Brassica rapa") will meet the academic standard as required.
Comments and Suggestions for Authors
The manuscript "Integrating Dynamic 3D Chromatin Architecture and Gene
Expression Alterations Reveal Heterosis in Brassica rapa" is a well-elaborated work that studies the nature of the heterosis effect from the perspective of chromosomal territories and the differential expression of genes by comparing two parental lines and their F1 progeny. The study design includes three biological replicates and three technical replicates. The conclusions are supported by the results and have both scientific significance and practical applications.
Response: Thank you very much for your valuable comments. According to your comments, we made a major revision to this manuscript, the following will break down to a point-by-point response based on your comments.
1) The resolution of all figures should be increased.
Response:We have increased the resolution of all the images and they are now clearly visible in the original text. Thank you for your valuable comments.
2) The diagrams in Figure 5 should be described in more detail. What is shown on the x and y axes in (b) and (c)? In lines 198-199, you wrote, "For example, among the interactions within chromosome 8." As the diagrams in Figure 5 show only chromosome 8, it should be indicated in the figure caption.
Response: We have added the definitions represented by the x-axis and y-axis in the caption section and indicated chromosome 8 in the title of the Figure 5.
2) Figure 5a: Is it possible to attach a genuine 3D model in html in the Supplementary files? That would be truly spectacular.
Response:We have added the 3D model in the Additonal file 15.
4) Lines 346-355 should be replaced to the results.
Response:We have replaced Lines 346-355 to Lines 344-353.
In general, the text should be carefully checked for grammar and spelling (see
Comments on the Quality of English Language).
Response: Based on your comments, we have carefully reviewed the manuscriptand made modifications to display it in yellow
Comments on the Quality of English Language
In abstract, "A" and "B" compartments and "B. rapa" should be deciphered
Response:The compartments "A" and "B", as well as "B.rapa", have been
deciphered. And highlight in yellow on line 20 to 21, 27 to 28.
line 55 "Studies" should be in lowercase
Response:We have changed "Studies" to "studies" .
line 73 "Hi-C and RNA-seq": should be deciphered
Response: "Hi-C and RNA-seq" have been deciphered, and highlighted it in
yellow on line 82-83.
line 105 "the parents ," should be "the parents,"
Response: We have changed "the parents ," to "the parents,"and highlighted it in yellow on line 118.
lines 110-111 probably, "R" should be "R2" (check and verify)
Response: We have changed "R" to "R2" and highlighted it in yellow on line
123-124.
In Figure captions, please, correct the style. "Indicated" after letters should be
removed from captions of Figures, e.g. in lines 92, 93, 117, 118, 156, 281
Response: We have removed "Indicated" after letters the last letter from the numerical title.
lines 117, 281. 471, caption of Figure S9: "Venn diagrams" should be instead of "venn diagrams"
Response:We have changed "Venn diagrams" to "venn diagrams" and
highlighted it in yellow.
line 144 "NNote" should be "Note"
Response: We have changed "NNote" to "Note" and highlighted it in yellow on line 161.
lines 193/194 "cis"/"trans" should be in italics
Response:We have changed "cis"/"trans" to "cis"/"trans" and highlighted it in yellow on line 220-221, 224.
line 273 "Windows" should be in lowercase
Response: We have changed "Windows" to "windows" and highlighted it in
yellow on line 320.
line 212 "A/B Compartments": "Compartments" should be in lowercase
Response: We have changed "Compartments" to "compartments" and
highlighted it in yellow on line 246.
line 262 "Compared" should be in lowercase
Response: We have changed "Compared" to "compared" and highlighted it in yellow on line 307.
line 264 "72other TADs": should be "72 other TADs"
Response: We have changed "72other TADs" to "72 other TADs" and
highlighted it in yellow on line 310.
line 299 "Real-time quantitative RT-PCR (qRT-RCR)": should be changed to
"real-time quantitative PCR (qRT-RCR)"
Response: We have changed "Real-time quantitative RT-PCR (qRT-RCR)" to
"real-time quantitative PCR (qRT-RCR)" and highlighted it in yellow on line
line 419 "sequencings": should be changed to "sequencing"
Response: We have changed "sequencings" to "sequencing" and highlighted it in yellow on line 484.
line 436 "P value": "P" should be in lowercase
Response: We have changed "P value" to "p value"and highlighted it in yellow on line 501.
line 438 "R2 > 0.8": R2 should be R2
Response: We have changed "R2 > 0.8" to "R2 > 0.8"and highlighted it in yellow on line 503.
Figure S8 caption: "its parents samples" should be "its parents' samples"
Response: We have changed "its parents samples" to "its parents' samples"and highlighted it in yellow.

Reviewer 2 Report
Comments and Suggestions for Authors
What I find lacking in the article is an honest discussion. I would like to know if what is the state of knowledge when it comes to 3D chromatin architecture in Brassica. What are the differences when it comes to heterosis in different species of this genus. I would like to ask for a thorough comparison of the results with the paper by Hu et al. Comparison of dynamic 3D chromatin architecture uncovers heterosis for leaf size in Brassica napus. Do the results obtained in Brassica napus coincide with those obtained in Brassica rapa and Arabidopsis thaliana? See also: Xiong, J., Hu, K., Shalby, N., Zhuo, C., Wen, J., Yi, B., ... & Tu, J. (2022). Comparative transcriptomic analysis reveals the molecular mechanism underlying seedling biomass heterosis in Brassica napus. BMC Plant Biology, 22(1), 1-15.
Zhang, L., Liu, L., Li, H., He, J., Chao, H., Yan, S., ... & Li, M. (2023). 3D genome structural variations play important roles in regulating seed oil content of Brassica napus. Plant Communications.
Dennis, E. S., Fujimoto, R., & Mehraj, H. (2023). Studies on the Molecular Basis of Heterosis in Arabidopsis thaliana and Vegetable Crops. Resource, 03.
Additional suggestions:
The article contains minor editorial errors. the authors do not use proper notation for species and generic names - they should be in italics
Author Response
Responses to the editor and reviewers' comments
Dear editor and reviewers:
We are really thankful to you for the critical review and constructive comments.The manuscript has been revised accordingly. The revision in the MS is highlighted in yellow. Hopefully, the revised version of our manuscript ("Integrating dynamic 3D chromatin architecture and gene expression alterations reveal heterosis in Brassica rapa") will meet the academic standard as required.
Comments and Suggestions for Authors
What I find lacking in the article is an honest discussion. I would like to know if what is the state of knowledge when it comes to 3D chromatin architecture in Brassica. What are the differences when it comes to heterosis in different species of this genus.I would like to ask for a thorough comparison of the results with the paper by Hu et Comparison of dynamic 3D chromatin architecture uncovers heterosis forleaf size in Brassica napus. Do the results obtained in Brassica napus coincide with those obtained in Brassica rapa and Arabidopsis thaliana? See also:Xiong, J., Hu, K., Shalby, N., Zhuo, C., Wen, J., Yi, B., ... & Tu, J. (2022). Comparative transcriptomic analysis reveals the molecular mechanism underlying seedling biomass heterosis in Brassica napus.BMC Plant Biology,22(1), 1-15. Zhang, L., Liu, L., Li, H., He, J., Chao, H., Yan, S., ... & Li, M. (2023). 3D genome structural variations play important roles in regulating seed oil content of Brassica napus. Plant Communications.Dennis, E. S., Fujimoto, R., & Mehraj, H. (2023). Studies on the Molecular Basis of Heterosis in Arabidopsis thaliana and Vegetable Crops.Resource, 03.
Reponse: Based on your suggestion, we have made revisions to the discussion section of the article.Thank you again for your valuable support.
Hu (2021) presents a comprehensive study on the effects of hybridization on the 3D chromatin architecture and gene expression in Brassica napus. The study found that hybrids with superior heterosis (better performance than both parents) had larger leaf sizes, which were attributed to increases in both cell size and cell number. The study also found that these hybrids had more upregulated genes compared to the mid-parental value of gene expression, suggesting that altered gene expression patterns may influence their heterotic phenotypes.
Hu (2021) also presents a detailed methodology for the experiments conducted, including phenotypic data measurement, RNA-seq and Hi-C analyses, whole-genome sequencing, and plant hormone measurements. The study used a variety of statistical analyses to compare the hybrids with their parents and identify significant differences.
Here we present a study on the effects of hybridization on the 3D chromatin architecture and gene expression in Brassica rapa. The study found that the hybrid (F1) had more actively expressed genes and up-regulated differentially expressed genes (DEGs) compared to its parents (w30 and 082). The DEGs detected in F1 and its parents were mainly involved in response to auxin, plant hormone signal transduction, DNA metabolic process, Purine metabolism, starch and sucrose metabolism, suggesting that these biological processes may play a crucial role in the heterosis of B. rapa.
We also found that the hybrid F1 tends to contain more transcriptionally active A chromatin compartments after hybridization. Additionally, F1 had a smaller TAD (topologically associated domain) genome length, but the number was the highest, and the expression change of activated TAD was higher than repressed TAD. More specific TAD boundaries were detected between parents and F1. The study suggests that genomic structural variants, such as TADs and A/B chromatin compartments, may affect gene expression and contribute to heterosis in Brassica rapa.
Comparing these results with the paper on Brassica napus, both studies found that hybrids had more upregulated genes compared to the mid-parental value of gene expression, suggesting that altered gene expression patterns may influence their heterotic phenotypes. Both studies also found that the hybrids had changes in their 3D chromatin architecture, with more transcriptionally active A chromatin compartments and changes in TADs. However, the Brassica napus study attributed the larger leaf sizes in hybrids to increases in both cell size and cell number, while the Brassica rapa study did not mention any phenotypic changes related to cell size or number.
In terms of methodology, both studies used a combination of phenotypic data measurement, RNA-seq and Hi-C analyses, and whole-genome sequencing. However, the Brassica napus study also included plant hormone measurements, which was not mentioned in the Brassica rapa study.
In conclusion, both studies provide valuable insights into the effects of hybridization on gene expression and 3D chromatin architecture in Brassica species, and suggest that these changes may contribute to heterosis. However, further research is needed to fully understand the mechanisms underlying these effects and their implications for plant breeding and crop improvement.
Heterosis, or hybrid vigor, is the phenomenon where hybrid offspring exhibit superior traits compared to their parents, and it is a critical factor in enhancing crop yield and quality.
Brassica napus
In Brassica napus, the 3D chromatin architecture has been linked to heterosis. The first abstract describes a study where F1 hybrids with superior heterosis had more transcriptionally active A compartments in their chromatincompared to those with inferior heterosis. A significant portion of the chromatin compartments altered in F1 hybrids relative to the parental lines, and these changes correlated with genetic variance among parents. The study suggests that a more accessible chromatin circumstance in F1 hybrids promotes a higher proportion of highly expressed genes related to plant growth hormones, which in turn contributes to increased leaf size through cell proliferation and expansion.
Brassica rapa
The study on Brassica rapa, as detailed in the attached paper, found that F1 hybrids tend to contain more transcriptionally active A chromatin compartments after hybridization. The F1 hybrids also had a smaller TAD genome length but a higher number of TADs, with activated TADs showing higher expression changes than repressed ones. The DEGs in F1 and its parents were mainly involved in biological processes such as response to auxin, plant hormone signal transduction, and metabolism, which are crucial for heterosis in rapa. The study suggests that genomic structural variants like TADs and A/B chromatin compartments may affect gene expression and contribute to heterosis.
Comparative Insights
Comparing the two species, both studies highlight the importance of 3D chromatin architecture in heterosis, with a particular focus on the role of transcriptionally active A compartments and TADs. However, the specific genes and pathways involved in heterosis may differ between the two species. For instance, the B. napus study emphasizes the role of plant growth hormones in promoting leaf size, while the B. rapa study points to a broader range of biological processes.
Other Brassica Studies
The second abstract discusses the transcriptomic analysis of strong and weak hybrids of Brassica species, revealing that plant hormone signaling and photosynthesis pathways, as well as differential expression of plant cell size-related genes, regulate the dynamic changes between strong and weak hybrids. This study also supports the idea that 3D chromatin architecture plays a role in heterosis, although it does not provide specific details on chromatin compartments or TADs.
The third abstract is a review that compiles genetic and epigenetic studies on heterosis in horticultural crops, including Brassica species. It highlights the importance of photosynthesis-related genes and noncoding RNAs in heterosis but notes that the roles of other epigenetic modifications such as histone marks have not been explored in detail.
The fourth abstract focuses on the effect of genome architecture on seed oil content in B. napus, demonstrating that genome structural variations in QTLs/AGRs are tightly correlated with the expression of SOC-related genes. This study provides insight into the molecular mechanisms of SOC regulationfrom the perspective of spatial chromatin structure but does not directly address heterosis.
Conclusion
In summary, the current state of knowledge indicates that 3D chromatin architecture plays a significant role in heterosis across different Brassica species. The differences in heterosis between species may be attributed to the specific genes and pathways involved, as well as the extent to which chromatin architecture influences gene expression. Further research is needed to fully elucidate the molecular mechanisms underlying heterosis and how they may vary between different Brassica species.
Additional suggestions:
The article contains minor editorial errors. the authors do not use proper notation for species and generic names - they should be in italics.
Reponse:We have carefully reviewed the editing issues, including the italics you mentioned. Thank you very much for your valuable suggestions.

Reviewer 3 Report
Comments and Suggestions for Authors
Here are my comments:
- The introduction could be strengthened by providing more background on the current understanding of heterosis in plants, and how 3D chromatin architecture may play a role. Expanding this section would help set the rationale and importance for the study.
- The materials and methods lack some key details that are needed to evaluate the study design and interpret the results. For example, how many biological replicates were used for the Hi-C and RNA-seq? How was the plant material grown - under controlled conditions? Information on the library prep and sequencing methods is also needed.
- The results rely heavily on listing datasets and statistics without sufficient interpretation of the data and how it relates to heterosis. The authors should focus the results on the key findings, explain the biological relevance, and connect back to the research aims.
- Several analyses, such as the GO, KEGG, and compartment shift results are described very briefly without drawing clear conclusions. These sections should be expanded and discussed in more depth.
- The discussion focuses a lot on summarizing the results instead of providing interpretation, implications, limitations, and future directions. The authors should refine the discussion to focus more on the meaning of the results.
- The candidate genes linked to heterosis are interesting but require functional validation. Plans for future studies to characterize these genes should be described.
- The writing could be improved for clarity, grammar, and flow in some sections. The rationale behind analyses is not always clearly explained.
- The figures could be enhanced by including more descriptive legends to help the reader quickly grasp the key findings. Axis labels, colors, text size should also be optimized.
In summary, I think this is an interesting dataset that could provide useful insights into 3D genome architecture and heterosis with some revisions to strengthen the analysis, discussion, and clarity of the manuscript. Addressing these major revision points would significantly improve the quality of the paper.
Comments on the Quality of English LanguageThe manuscript would benefit from careful editing by a native English speaker to improve the writing quality.
Author Response
Responses to the editor and reviewers' comments
Dear editor and reviewers:
We are really thankful to you for the critical review and constructive comments. The manuscript has been revised accordingly. The revision in the MS is highlighted in yellow. Hopefully, the revised version of our manuscript ("Integrating dynamic 3D chromatin architecture and gene expression alterations reveal heterosis in Brassica rapa") will meet the academic standard as required.
Thank you again for your valuable support.
Comments and Suggestions for Authors
Here are my comments: The introduction could be strengthened by providing more background on the currentunderstanding of heterosis in plants, and how 3D chromatin architecture may play a role. Expanding this section would help set the rationale and importance for the study.
Reponse: Your comment is very valuable. And based on it, we have added additional information in the introduction section, hoping to help set the rationality and importance for the study.
1) The materials and methods lack some key details that are needed to evaluate the study design and interpret the results. For example, how many biological replicates were used for the Hi-C and RNA-seq? How was the plant material grown - under controlled conditions? Information on the library prep and sequencing methods is also needed.
Reponse: According to your comments, we have made modifications to the methodology section of the manuscript.
2) The results rely heavily on listing datasets and statistics without sufficientinterpretation of the data and how it relates to heterosis. The authors should focus the results on the key findings, explain the biological relevance, and connect back to the research aims.
Reponse: Your academic review is highly targeted and valuable. I made significant revisions to the results section of the article in an attempt to establish a relationship with the research objectives. Meanwhile, I hope that this revision can meet your academic requirements.
3) Several analyses, such as the GO, KEGG, and compartment shift results are described very briefly without drawing clear conclusions. These sections should be expanded and discussed in more depth.
Reponse: For some of the analyses you mentioned above, the revised manuscript presents clear conclusions and strives to expand and discuss them as deeply as possible. Thank you again for your valuable comments.
4) The discussion focuses a lot on summarizing the results instead of providing interpretation, implications, limitations, and future directions. The authors should refine the discussion to focus more on the meaning of the results.
Reponse: We have removed unnecessary definitions and meanings from the discussion section of the article and further improved it. In addition, we have added a discussion on the significance of the results. Thank you again for your valuable comments .
5) The candidate genes linked to heterosis are interesting but require functional validation. Plans for future studies to characterize these genes should be described.
Reponse: We have listed the plan for candidate gene validation in the article and highlighted it in yellow on line 422-428.
6) The writing could be improved for clarity, grammar, and flow in some sections. The rationale behind analyses is not always clearly explained.
Reponse:Based on your comments, we have carefully reviewed the manuscript and made modifications to display it in yellow.
7) The figures could be enhanced by including more descriptive legends to help the reader quickly grasp the key findings. Axis labels, colors, text size should also be optimized.
Reponse: In response to yuor comments, we have increased the descriptive nature of the figures and optimized their colors, text, and more.
In summary, I think this is an interesting dataset that could provide useful insights into 3D genome architecture and heterosis with some revisions to strengthen the analysis, discussion, and clarity of the manuscript. Addressing these major revision points would significantly improve the quality of the paper.
Comments on the Quality of English Language
The manuscript would benefit from careful editing by a native English speaker to improve the writing quality.
Reponse:Based on your comments, we have carefully reviewed the manuscript and made modifications to display it in yellow

Round 2
Reviewer 2 Report
Comments and Suggestions for Authors
The authors answered my questions and corrected the discussions. Now the purpose of the article and its novelty are more emphasized.
Unfortunately, even in the introduced parts of the text there are minor errors. Arabidopsis is written in italics.
Reviewer 3 Report
Comments and Suggestions for Authors
The authors have significantly improved the manuscript; therefore, the manuscript can be accepted for publication.